# Online Recommendation Systems: Factors Influencing Use in E-Commerce

**Juan-Pedro Cabrera-Sánchez [1]**, **Iviane Ramos-de-Luna [2]**, **Elena Carvajal-Trujillo [3]**
**and Ángel F. Villarejo-Ramos [1,\*]**

[1]   Business Administration and Marketing Department, Universidad de Sevilla, 41018 Sevilla, Spain;
      jcabrera10@us.es
[2]   Economics and Business Studies Department, Universitat Oberta de Catalunya, 08035 Barcelona, Spain;
      iramosde@uoc.edu
[3]   Business Administration and Marketing Department, Universidad de Huelva, 21071 Huelva, Spain;
      carvajal.trujillo@dem.uhu.es
\*   Correspondence: curro@us.es; Tel.: +34-954-56-00-24

**Abstract:** The increasing use of artificial intelligence (AI) to understand purchasing behavior has led to the development of recommendation systems in e-commerce platforms used as an influential element in the purchase decision process. This paper intends to ascertain what factors affect consumers' adoption and use of online purchases recommendation systems. In order to achieve this objective, the Unified Theory of Adoption and Use of Technology (UTAUT 2) is extended with two variables that act as an inhibiting or positive influence on intention to use: technology fear and trust. The structural model was assessed using partial least squares (PLS) with an adequate global adjustment on a sample of 448 users of online recommendation systems. Among the results, it's highlighted the importance of the inhibiting role of technology fear and the importance that users attach to the level of perceived trust in the recommendation system are highlighted. The performance expectancy and hedonic motivations have the greatest influence on intention to use these systems. Based on the results, this work provides a relevant recommendation to companies for the design of their e-commerce platforms and the implementation of online purchase recommendation systems.

**Keywords:** recommendation system; artificial intelligence; e-commerce; technology fear; trust

## 1. Introduction

Being able to make decisions based on data is an unquestionable advantage for companies. New technologies such as big data and artificial intelligence (AI) have radically changed data analytics [1]. These technologies represent a revolution in how companies can obtain a great quantity of valuable data and how they analyze them. They enable them to perform a predictive [2], descriptive, and even prescriptive analysis of their market and the environment [3]. Furthermore, these quasi-real-time analyses represent a great competitive advantage for companies [4] and challenge them to develop sustainable competitive advantages [5].

The technology adoption by companies and consumers is decisive for its success. Therefore, if the factors that affect its adoption and behavioral intention are known, implementation in companies and acceptance by consumers/users could be better and more easily managed. From the classical theories of acceptance of technology—the Technology Acceptance Model (TAM) [6] and the Theory of Planned Behavior (TPB) [7]—the model proposed by the Unified Theory of Adoption and Use of Technology (UTAUT) [8] brings together the most outstanding contributions of the previous models. This model adequately explains the adoption of technology by companies but had to be revised and expanded in

order to explain consumer adoption of technology, giving rise to UTAUT 2 [9]. This model provides excellent results when studying consumer acceptance and use of technology, but the question remains whether there are other variables that extend and improve the UTAUT 2 model.

Recently, big data has made great advances due to AI algorithms, which at the same time have benefited from the development of big data technology [10]. Although there are more benefits than disadvantages, there are still certain barriers to its acceptance and use: ignorance, technological fear, distrust, resistance to change, or the limitations of the technology in itself [11].

Most of the literature on big data and AI focuses, fundamentally, on technical aspects related to the big data ecosystem and new AI algorithms [10,12–14], application development, statistical modeling and experimentation [4], and study cases on data mining and data analytics [15]. However, few works integrate the use of big data or AI tools in a business management system. On the one hand, the big data was investigated in different contexts: the behavioral intention of big data analytics [16]; the use of big data to obtain online consumer reviews (OCR), which will benefit companies and consumers in an e-commerce context [17]; big data technology adoption [18,19]; and the effect of resistance to the use of big data techniques in companies [20]. On the other hand, research into AI applications and implementation only exists in specific sectors, such as education [21], health [22], consumer privacy [23], and social networks [24].

Purchase recommendation systems use the information from online buyers' digital activity to make suggestions and recommendations to consumers when they start the conversion process in an e-commerce platform. According to Hong et al. [25] and Fu et al. [26], the perception of review helpfulness is an important step in the online shopping decision-making process. This study intended to ascertain what factors affect the adoption and use of online purchase recommendation systems. Consumers are extremely vulnerable to biased personalized online recommendations [27].

Section 2 of this paper addresses the theoretical justification of the relations of the proposed causal model. Section 3 outlines the methodology used in the research, and in Section 4 we describe and analyze the results obtained after applying the model in the sample observed. The paper finishes with the main theoretical and practical conclusions, as well as the study's limitations.

## 2. Theoretical Background and Hypotheses

The integration characteristics of the UTAUT model [8] make it a suitable model to measure the level of acceptance and use of tools developed from artificial intelligence (AI). The extended version, UTAUT 2 [9], oriented towards explaining the use of technologies in consumer markets, will make it possible to understand the acceptance and the use of certain applications developed from AI in online purchase situations. This paper focuses on the information that recommendation systems provide to online buyers in e-commerce.

The reference terms of the UTAUT 2 model consider precedents of behavioral intention of use of technology: (1) performance expectancy (PE), defined as the degree to which using a technology offers benefits (usefulness) in performing certain activities; (2) effort expectancy (EE), which measures the degree of facility (ease of use) associated with the use of the technology; (3) social influence (SI) or how consumers perceive that friends and family think that they should use a technology; (4) facilitating conditions (FC), consumers' perceptions of which resources and support are available to engage in a behavior; (5) hedonic motivations (HM), measured through perceived enjoyment as a determinant of technology use; (6) price value (PV), the value perceived by the consumer in relation to the price paid for using the technology; and (7) habit (HT), measured by the habitual and natural use of the technology.

Habit and facilitating conditions also directly influence the use (Use) of the technology. Furthermore, behavioral intention of use of technology (BI) positively influences the use.

Arenas-Gaitán et al. [28] indicate that the value of UTAUT 2 lies in its ability to identify the principal determinants of adoption. The effect of different moderating variables can be included and considered in the model. In our case, latent variables, technology fear (TF) and trust (TR), are included in an attempt to extend the model and to improve its explanatory capacity.

Different hypotheses based on the extension of the UTAUT 2 model for the case of acceptance and use of purchase recommendation systems in e-commerce users are proposed.

Performance expectancy is one of the most influential variables on behavioral intention. Several papers [18,29,30], besides the original paper itself [8], support this positive relation. Therefore, we formed a hypothesis:

**Hypothesis 1 (H1).** *Performance expectancy positively influences the behavioral intention of recommendation systems in e-commerce.*

Other researchers [29–33] reinforce the sense and the weight of the effect that effort expectancy has on behavioral intention. Thus, we formed the second hypothesis of the model:

**Hypothesis 2 (H2).** *Effort expectancy positively influences the behavioral intention of recommendation systems in e-commerce.*

In the original model of Venkatesh et al. [8] and the extended model UTAUT 2 [9], social influence measures the extent to which users are influenced by what others (friends, family) think about the use of technology. Some previous papers in the same online purchase context show that social influence is a very strong antecedent of behavioral intention [20,34–36] and adoption of an IT platform [37]. Thus, we formed a new hypothesis:

**Hypothesis 3 (H3).** *Social influence positively influences the behavioral intention of recommendation systems in e-commerce.*

Hedonic motivations are the pleasure and/or the enjoyment obtained through using technology [9]. This enjoyment is a relevant predictor of the behavioral intention of the technology [38–40]. Other researchers with similar technologies, such as m-commerce [41] or tourist geolocation [42], show hedonic motivation as one of the main influences on the behavioral intention of these applications by users. Hence, we formed the fourth hypothesis of our research:

**Hypothesis 4 (H4).** *Hedonic motivation positively influences the behavioral intention of recommendation systems in e-commerce.*

To extend the consumer context of the acceptance model, UTAUT 2 adds a new variable that measures perceived value beyond the benefit for consumers who use the technology. This variable, price value, compares the cost for consumers with the benefit of using it [9]. Although some papers about free access mobile technologies [42] cannot show the positive influence on behavioral intention, we formed an original hypothesis:

**Hypothesis 5 (H5).** *Price value positively influences the behavioral intention of recommendation systems in e-commerce.*

Facilitating conditions represent easy access to the resources needed to use a new technology and the subsequent support [8]. More recent studies have verified this positive effect on behavioral intention [43–45].

According to the Theory of Planned Behavior [7], we can test how facilitating conditions positively affect the use of a new technology. Various subsequent papers [18,29,31,43,46,47] show this relationship. Accordingly, we formed the following hypotheses:

**Hypothesis 6a (H6a).** *Facilitating conditions positively influence the behavioral intention of recommendation systems in e-commerce.*

**Hypothesis 6b (H6b).** *Facilitating conditions positively influence the use of recommendation systems in e-commerce.*

Habit is the usual conduct of a person because of a learnt behavior [48]. In the UTAUT 2 model [9], the authors denote habit as past expertise and outline how this expertise represents the reason to use new technologies [49]. Previous studies demonstrated the influence of habit on the behavioral intention of use of a technology [50,51], and habit cannot only affect intention to use but also use of this technology [42,48,52]. Therefore, two hypotheses were formed:

**Hypothesis 7a (H7a).** *Habit positively influences the behavioral intention of recommendations system in e-commerce.*

**Hypothesis 7b (H7b).** *Habit positively influences the use of recommendation systems in e-commerce.*

One of the variables in the extended model of UTAUT 2 is technology fear. Initially, this concept was developed as technology anxiety in works such as Guo et al. [53], Niemelä-Nyrhinen [54], and Venkatesh [55]. These researchers demonstrate that the greater the level of novelty of the technology, the bigger the anxiety effect. As Martínez-Córcoles et al. [56] indicate, adoption of new technologies is largely influenced by technology fear and discomfort. Fear has an important effect on the behavioral intention to use when the user has not used the technology before [57]. The proliferation of new technologies leads consumers to adopt them quickly, which leads to paired emotional and cognitive reactions, with fear appearing among consumers [58]. Technology fear, different from IT anxiety, is considered close to technophobia including aversive behavioral, affective, and attitudinal responses to new technologies [56].

The fear of the consequences of using a new technology can hinder behavioral intention, and this situation causes insecurity and a feeling of intimidation. Furthermore, fear is an important barrier for intention to use, as Heinssen Jr. et al. [59] show with the Computer Anxiety Rating Scale (CARS) in the dimension that is called fear. Thus, the following hypothesis was formed:

**Hypothesis 8 (H8).** *Technology fear negatively influences the behavioral intention of recommendation systems in e-commerce.*

Trust reflects a user's security about meeting expectations vis-à-vis the behavior of the other party in the relationship [60,61]. The user trusts the e-services provider to perform its tasks, comply with the service promise, and maximize user profit [62,63]. Trust has been one of the strongest predictors of the behavioral intention of e-commerce [46]. As Zhou [64] indicates, the user hopes to obtain a benefit by using technology. Therefore, it can positively and directly affect users' behavioral intention based on trust in this technology system. Therefore, the following hypothesis was formed:

**Hypothesis 9 (H9).** *Trust positively influences the behavioral intention of recommendation systems in e-commerce.*

The main technology acceptance models (TAM [6], UTAUT [8], and UTAUT 2 [9]) show a direct relationship between behavioral intention and technology usage. This influence has been demonstrated in similar contexts to an AI application, such as internet banking adoption [36], online flight purchases [65], acceptance of an electronic document management system [66], and ERP (Enterprise Resource Planning) adoption [29]. Therefore, we formed as a hypothesis:

**Hypothesis 10 (H10).** *The behavioral intention of recommendation systems in e-commerce positively affects the use.*

Figure 1 shows the proposed model of acceptance and use of recommendation systems in e-commerce as an AI application for users.

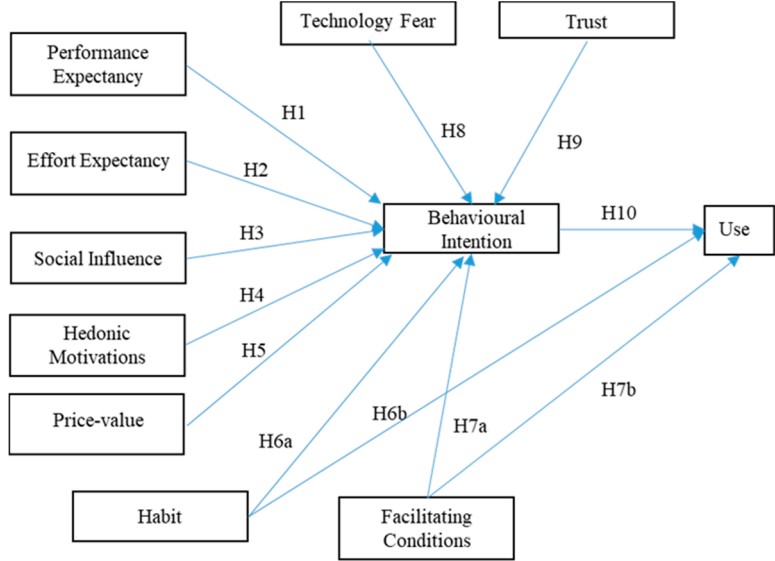

**Figure 1.** Proposed model.

## 3. Research Methodology

### 3.1. Sample Description

The sample drawn for this research came from individuals who answered an online questionnaire spread across social networks and mailing lists. Both circulation methods mainly reached under and postgraduate students since the messages originated from the university environment. The data were collected in March 2019, and most respondents were university graduates and postgraduates (42.6%). Finally, the selected sample consisted of 448 questionnaires from individuals who stated that they use online recommendation systems when faced with purchasing decisions. Analysis of sociodemographic variables in the sample indicated that 53.1% of participants are female, the average age is 27.1 years old, and 71.6% are single. Only 23.2% are hired workers, and 64.9% are students. Furthermore, 68.75% of participants live at home and have a monthly income of more than €1500. More than half of the sampled (50.7%) participants live in cities with more than 100,000 people.

This questionnaire was pre-tested with several users and with expert researchers who completed the questionnaire and provided feedback about the questions, testing content validity.

### 3.2. Measurement Scales

The scales were adapted from the original UTAUT model of Venkatesh et al. [8], the extended UTAUT 2 model [9], and from Davis' [67] TAM. The scale items were adapted to online recommendations systems (ORSs) in e-commerce as an application of AI, tested in our research. The proposed model adds scale of technology fear adapted from Heinssen et al. [59] and the scales of trust adapted from Pavlou and Gefen [68]. All scales were measured using a 7-point Likert scale.

In the questionnaire, we used the concept of online recommendation systems (ORSs) as an AI tool/application. In the heading of the questionnaire used we include the following explanation of what an ORS is: Every time that we buy something, for example, on Amazon, it makes additional products recommendations according to our profile and previous purchases, or if you are looking for something or a place, your phone recommends where to go or what to do.

The sample used was taken from a higher-ranking study that researched the acceptance and use of other IA apps, such as anti-spam filters, geolocation tools, and facial and voice recognition, among others.

### 3.3. Statistical Tools

To test the proposed structural model, we used partial least squares–structural equation modeling (PLS-SEM) before analyzing the reliability and the validity of the measurement scales [69–71], specifically the statistical software Smart PLS 3.2.3 [72].

The recommendations of Kock [73] and Kock and Lynn [74] suggest that we first check the absence of measure bias error or common method bias (CMB). To do so, we added various items not known to have been researched and that were not in the proposed structural model. This new latent variable is called the CMB variable, which acts as the dependent variable of the others in the model. All variance inflation factors (VIFs) extracted by this method must be lower than 3.3 to confirm that the sample has no CMB. Table 1 shows compliance with these requirements.

**Table 1.** Variance inflation factors (VIFs) extracted from the variables to test the common method bias (CMB).

|  | CMB Variable |
|---|---|
| Behavioral intention | 1687 |
| Effort expectancy | 1731 |
| Facilitating conditions | 1858 |
| Habit | 1348 |
| Hedonic motivations | 1652 |
| Performance expectancy | 1852 |
| Price value | 1321 |
| Social influence | 1515 |
| Technology fear | 1259 |
| Trust | 1360 |
| Use | 1046 |

## 4. Results

Structural equation modeling (SEM), as in the case of PLS analysis, firstly tries to ensure the reliability and validity of the measurement scales.

To analyze each item's individual reliability, its loadings must be observed. They must be above the recommended minimum level (0.7) when using variables measured in the B mode [75,76]. All the items, except FC4, TF1, TF2, TF4, and TF8, met this requirement. Therefore, we decided to eliminate these indicators from the scale of technology. Table 2 shows items and loadings over 0.7 in all cases.

We analyzed the reliability of the constructs with two indicators: composite reliability and Cronbach's alpha. The values obtained were over the 0.7 suggested [77]. To analyze the convergent validity, we used the average variance extracted (AVE). The results were over the 0.5 recommended [62]. These results can be seen in Table 3.

The discriminant validity of the measurement model was successfully carried out with a double test: (1) Fornell–Larcker's test, which compares the square roots of the AVE (shown in the diagonal in bold) with the correlations of each variable (shown in rows and columns). The first value must be greater than those in their respective rows and columns [78]; and (2) the heterotrait–monotrait ratio (HTMT), where all the values of the correlations between variables must be less than 0.9 and preferably less than 0.8 [75]. Table 4 shows both results.

**Table 2.** Measurement scales and loadings.

| Construct | Items | Loadings |
|---|---|---|
| Effort expectancy—EE | EE1: I find it easy to learn to use ORSs. | 0.877 |
| | EE2: My interaction with ORSs is clear and comprehensible. | 0.901 |
| | EE3: I find it easy to use ORS. | 0.909 |
| | EE4: I believe that learning applications that use ORSs is easy for me. | 0.873 |
| Performance expectancy—PE | PE1: I believe ORSs are useful for me in my day-to day life. | 0.812 |
| | PE2: I believe that with ORSs my possibilities of achieving things that are important for me increase. | 0.830 |
| | PE3: I believe ORSs help me to carry out my tasks more quickly. | 0.793 |
| | PE4: I believe ORSs improve my performance. | 0.830 |
| Social influence—SI | SI1: People who I care about think I should use ORSs. | 0.914 |
| | SI2: People who influence my behavior think I should use ORS applications. | 0.955 |
| | SI3: People whose opinion I value and consider believe I should use ORSs. | 0.939 |
| Hedonic motivations—HM | HM1: Using ORSs is fun. | 0.924 |
| | HM2: I enjoy using ORSs. | 0.957 |
| | HM3: Using ORSs is very entertaining. | 0.937 |
| Price value—PV | PV1: ORSs are reasonably priced. | 0.815 |
| | PV2: ORSs are worth what they cost. | 0.905 |
| | PV3: At the current price ORSs give good value. | 0.931 |
| Habit—HT | HT1: The use of ORSs has become a habit for me. | 0.858 |
| | HT2: I am an ORS addict. | 0.836 |
| | HT3: I must use ORSs. | 0.829 |
| | HT4: Using ORSs has become something natural for me. | 0.869 |
| Facilitating conditions—FC | FC1: I have the necessary resources to use ORSs. FC2: I have the necessary knowledge to use AI applications. FC3: AI is compatible with other applications I use. FC4: When I have trouble using AI applications, I can get help. | 0.851 |
| | FC2: I have the necessary knowledge to use ORSs. | 0.858 |
| | FC3: ORSs are compatible with other applications I use. | 0.864 |
| Technology Fear—TF | TF3: I feel afraid when working with ORSs. | 0.753 |
| | TF5: I feel anxiety when working with ORSs. | 0.875 |
| | TF6: I feel insecure about my capacity to understand ORSs. | 0.813 |
| | TF7: I have avoided ORSs because I am not familiar with them and, to a certain extent, they intimidate me. | 0.884 |

**Table 2.** *Cont.*

| Construct | Items | Loadings |
|---|---|---|
| Trust—TR | TR1: One can trust ORSs. | 0.757 |
| | TR2: ORSs fulfil what they promise. | 0.812 |
| | TR3: ORSs deal with satisfying the user. | 0.836 |
| Behavioral intention—BI | BI1: I intend to use ORSs in the near future. | 0.871 |
| | BI2: I will always try to use ORSs in my daily life. | 0.907 |
| | BI3: I plan to use ORSs frequently. | 0.932 |

**Table 3.** Composite reliability and convergent validity.

| Construct | Cronbach's Alpha | rho_A | Composite Reliability | Average Variance Extracted (AVE) |
|---|---|---|---|---|
| Behavioral Intention | 0.887 | 0.888 | 0.930 | 0.817 |
| Effort Expectancy | 0.913 | 0.928 | 0.939 | 0.793 |
| Facilitating Conditions | 0.821 | 0.826 | 0.893 | 0.736 |
| Habit | 0.870 | 0.871 | 0.911 | 0.720 |
| Hedonic Motivation | 0.933 | 0.934 | 0.957 | 0.882 |
| Performance Expectancy | 0.834 | 0.836 | 0.889 | 0.667 |
| Price Value | 0.862 | 0.903 | 0.915 | 0.783 |
| Social Influence | 0.929 | 0.931 | 0.955 | 0.876 |
| Technology Fear | 0.857 | 0.932 | 0.900 | 0.694 |
| Trust | 0.730 | 0.758 | 0.844 | 0.644 |
| Use | 1.000 | 1.000 | 1.000 | 1.000 |

We evaluated the structural model to test the hypotheses proposed and analyze the path coefficients. We carried out bootstrapping with 10,000 sub-samples to check the statistical significance of each of the coefficients or paths (see Table 5). We used the SRMR criterion (Standardized Root Mean Square Residual) to assess the model's goodness of fit [79]. The value obtained was 0.055, less than the 0.08 proposed by [75], suggesting a good fit of the complete model.

The model has a higher explanatory power than the minimum level of 10% recommended by [80], measured by the explained variance of the endogenous variables ($R^2$).

The model results in Table 5 show both the verified hypotheses and the size of their effect. We must emphasize that although the $R^2$ of the Intention to use is well above the minimum recommended value (0.1) [80], the same does not apply to Use, the $R^2$ of which is lower. It is curious to see how the model explains the intention to use but not the use that is made. Seven of the 12 relationships formulated are significant, of which only the results for expectations–intention to use ratio exceed the minimum 0.2 recommended by Chin [81]. In all cases where the ratios were significant, the effect size (f2) was between 0.015 and 0.35 [81], except in scenarios 7b, 9, and 10, which were below it. However, the model has a very good fit as the SRMR obtained is only 0.055, well below 0.08 [82].

**Table 4.** Discriminant validity. Fornell–Larcker criterion (below the main diagonal) and heterotrait–monotrait ratio (HTMT) (above the main diagonal).

| | BI | EE | FC | HT | HM | PE | PV | SI | TF | TR | USE |
|---|---|---|---|---|---|---|---|---|---|---|---|
| **BI** | **0.904** | 0.384 | 0.386 | 0.394 | 0.526 | 0.597 | 0.379 | 0.408 | 0.251 | 0.456 | 0.156 |
| **EE** | 0.351 | **0.89** | 0.702 | 0.351 | 0.456 | 0.362 | 0.247 | 0.267 | 0.342 | 0.301 | 0.106 |
| **FC** | 0.331 | 0.604 | **0.858** | 0.43 | 0.413 | 0.3 | 0.411 | 0.159 | 0.452 | 0.383 | 0.108 |
| **HT** | 0.347 | 0.317 | 0.365 | **0.848** | 0.386 | 0.363 | 0.299 | 0.343 | 0.099 | 0.298 | 0.149 |
| **HM** | 0.479 | 0.423 | 0.366 | 0.348 | **0.939** | 0.556 | 0.401 | 0.396 | 0.192 | 0.373 | 0.07 |
| **PE** | 0.516 | 0.327 | 0.255 | 0.312 | 0.494 | **0.816** | 0.377 | 0.598 | 0.101 | 0.484 | 0.088 |
| **PV** | 0.338 | 0.214 | 0.339 | 0.26 | 0.363 | 0.325 | **0.885** | 0.285 | 0.14 | 0.408 | 0.051 |
| **SI** | 0.371 | 0.249 | 0.138 | 0.308 | 0.367 | 0.523 | 0.255 | **0.936** | 0.043 | 0.333 | 0.051 |
| **TF** | −0.236 | −0.306 | −0.381 | −0.034 | −0.178 | −0.088 | −0.127 | 0.014 | **0.833** | 0.272 | 0.067 |
| **TR** | 0.381 | 0.255 | 0.31 | 0.244 | 0.315 | 0.39 | 0.334 | 0.277 | −0.226 | **0.803** | 0.027 |
| **USE** | 0.148 | 0.101 | 0.099 | 0.14 | 0.068 | 0.082 | 0.05 | 0.05 | −0.032 | −0.02 | **1** |

Note: The values on the diagonal must be greater than the quantities in each column. If this is the case is in bold.

**Table 5.** Structural model estimates (path coefficients).

|  | **Estimates** | **P Values** | **$R^2$** | **f2** | **Validation** |
|---|---|---|---|---|---|
| Hypothesis 1: EE -> BI | 0.048 | n.s. |  | 0.002 | Not Supported |
| Hypothesis 2: PE -> BI | 0.258 *** | 0.000 |  | 0.064 | Supported |
| Hypothesis 3: SI -> BI | 0.076 | n.s. |  | 0.006 | Not Supported |
| Hypothesis 4: HM -> BI | 0.176 ** | 0.001 |  | 0.032 | Supported |
| Hypothesis 5: PV -> BI | 0.076 | n.s. |  | 0.007 | Not Supported |
| Hypothesis 6a: FC -> BI | 0.014 | n.s. |  | 0.000 | Not supported |
| Hypothesis 6b: FC -> USE | 0.030 | n.s. |  | 0.001 | Not Supported |
| Hypothesis 7a: HT -> BI | 0.111 * | 0.011 |  | 0.016 | Supported |
| Hypothesis 7b: HT -> USE | 0.092 * | 0.046 |  | 0.007 | Supported |
| Hypothesis 8: TF -> BI | −0.126 ** | 0.002 |  | 0.021 | Supported |
| Hypothesis 9: TR -> BI | 0.107 * | 0.017 |  | 0.014 | Supported |
| Hypothesis 10: BI -> USE | 0.106* | 0.027 |  | 0.010 | Supported |
| Behavioral Intention |  |  | 0.401 |  |  |
| Use |  |  | 0.032 |  |  |
| SRMR |  |  |  |  | 0.055 |

Note: *** $p < 0.001$, ** $p < 0.01$, * $p < 0.05$. (bootstrap with 10,000 sub-samples). n.s. (not significative).

We also calculated Stone–Geisser's $Q^2$ to evaluate the predictive capacity of the model, and by obtaining all $Q^2$ values greater than 0 we can conclude that it has a relevant predictive capacity [76] (see Table 6).

**Table 6.** Predictive summary of the latent variables.

|  | **RMSE** | **MAE** | **$Q^2$_Predict** |
|---|---|---|---|
| Behavioral Intention | 0.803 | 0.616 | 0.363 |
| Use | 0.999 | 0.877 | 0.009 |

## 5. Conclusions and Limitations

### 5.1. Theoretical Implications

This study extended the UTAUT model, adding two new explanatory variables: (1) technology fear—in this case, of an AI-based ORS application—and (2) trust in technology. We intended to increase the predictive value of the original model's explanation and to find out how factors (positively and negatively) influence the behavioral intention and use of a recommendation system in online purchasing.

The results of the research show that the behavioral intention of the recommendation systems receive positive and relevant influences of variables such as: (1) performance expectancy, the perception of obtaining good results when using the recommended information in shopping decisions, coinciding with the results of the work of Lee and Song [33] about e-government and the paper

of Yu [30] about the behavioral intention of internet banking; (2) the positive influence that hedonic motivations have on using recommendation systems, as stated by Zhang et al. [41] on online purchasing in the same context, and tourist recommendation [42] or adopting augmented reality smart glasses (ARSGs) [83]; (3) positive influence of habit, measured by the habitual use of these systems [52]; and (4) trust in the usefulness of the information provided by the recommenders [61].

Regarding the rest of the variables of the original UTAUT 2 model [9], we highlight an insignificant influence on the behavioral intention of recommendation systems of effort expectancy (the AI application's level of ease of use), social influence (what others consider appropriate to use), perceived price value (the perceived value in relation to the cost), and the facilitating conditions, considered as the ease of access to the application. The variable habit shows a significant favorable effect on use of the recommendation systems. However, the positive relationship between facilitating conditions and use was not supported.

With respect to the model's extended variables, we verified the significant influence of trust on recommendation systems concerning behavioral intention, already shown in papers in the digital environment [46,64], and it is a facilitator of use of this AI application. Technology fear displays a negative and significant effect on the intention of use of these recommendation systems, in line with the work of Gelbrich and Sattler [57].

In light of the results, and seeing the level of acceptance of the hypotheses proposed, we consider that extending UTAUT 2 with the fear inhibiting factor and the trust facilitator factor is an explanatory model of the intention and use by consumers in online purchasing recommendation systems.

## 5.2. Practical Implications

With regard to the professional implications, it should be noted that, as in previous works [20,29], this research indicates for some of the variables, such as the facilitating conditions and the effort expectancy, that as the level of digitalization of users and of the linked tools advances, online buyers/users cease to perceive the difficulty of use of these applications. They get used to these technologies and expect more perceived benefit than difficulty in their use [18]. This is in consonance with the strong effect exerted by performance expectancy. Furthermore, users consider that they have the resources and capacities to use them, particularly when it is a question of, as in our case, young buyers on average.

The fact that the two variables that have a greater influence on the behavioral intention of recommendation systems are performance expectancy and hedonic motivations leads us to recommend that the developers of these systems try to adapt them to the benefit expected by the online buyer, both at the efficiency level in the recommendations using big data analytics [84] given in the purchasing process and at the level of design and usability, to attain a greater enjoyment when these systems appear in the online purchasing process. In addition, like Xu et al. [85] suggest, product managers can draw real-time information regarding customers' recommendations and use this information to redesign new products.

With respect to trust and technology fear, factors that influence intention to use recommendation systems, campaigns should be carried out to convey the sensation of safety and trust in the acquisition and use of consumers' data to provide useful information in the online purchasing processes.

## 5.3. Limitations

Just including two variables to extend the UTAUT 2 model can result in a biased view, as the effect of other possible constructs, such as perceived risk or privacy conditions, was not considered. These constructs are closely linked to the acceptance of systems that use digital fingerprint-based data to provide information of interest, such as the AI-based recommendation systems that process data collected via big data tools.

Secondly, new moderator variables other than those of the original UTAUT ought to be explored to evaluate possible new effects not considered before. These could allow us to establish consumer behavior differences and define possible market segments.

Thirdly, the sample size will allow us in the future to establish behavioral differences between groups, which we can analyze via a posteriori segmentation technique, such as POS-PLS (Prediction-oriented Segmentation-Partial Least Squares).

Lastly, despite having intended to use the online collection method to broaden the sample's spectrum of representativeness, the sample had a certain bias in the low average age, high proportion of students, and high average income among the participants.

**Author Contributions:** Individual contributions by authors: Conceptualization, J.-P.C.-S. and Á.F.V.-R.; methodology, J.-P.C.-S., E.C.-T. and Á.F.V.-R.; software, J.-P.C.-S.; validation, J.-P.C.-S., Á.F.V.-R. and I.R.-d.-L.; formal analysis, J.-P.C.-S. and I.R.-d.-L.; investigation, J.-P.C.-S. and Á.F.V.-R.; resources, J.-P.C.-S., E.C.-T. and Á.F.V.-R.; data curation, J.-P.C.-S.; writing—original draft preparation, J.-P.C.-S., E.C.-T. and Á.F.V.-R.; writing—review and editing Á.F.V.-R. and I.R.-d.-L.; supervision, Á.F.V.-R.; project administration, Á.F.V.-R.. All authors have read and agreed to the published version of the manuscript.

**Funding:** This research received no external funding.

**Acknowledgments:** We are grateful for the technical support given by the Research Group MAD- Analytic and Digital Marketing of University of Seville and the recommendations received from Liébana-Cabanillas, Ph. D of the University of Granada.

**Conflicts of Interest:** The authors declare no conflict of interest.

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
