# Peer review of "Online Recommendation Systems: Factors Influencing Use in E-Commerce"

_sustainability, doi:10.3390/su12218888_

Round 1

Reviewer 1 Report

The study and overall report you provide is strong and certainly consistent with most quality research into consumer behaviour, measured according to the seven variables you outline as key to behavioural intention and the three variables affecting consumer use in general.

The relationships you map out and subsequently measure are highly significant. This work provides a generally useful set of relationships regarding the design and implementation of Online Recommendation Systems in order to improve e-commerce both socially and economically.

My main concern is really just the current edit, which needs more English language based changes. The changes required are in terms of both the syntax and sentence construction. The sentences are too complex, beginning with the title and syntactically misconstructed. Minor but significant changes are needed. 

Author Response

Firstly, thank you for your effort in reviewing our paper and, thank you for your kind words. No action is required.

To improve methodology, in the second paragraph of the 3.2 Epigraph we have included in the text of the revised manuscript:

"In the heading of the questionnaire used we include the following explanation of what an ORS is: Every time that we buy something, for example, in Amazon, it makes additional products recommendations according to our profile and previous purchases; or if you are looking for something or a place, your phone recommends where to go or what to do."

Furthermore, in order to ensure the predictive capability of the model we have calculated Stone-Geisser's Q2 and added the following paragraph at the end of section 4 (Results):

"We have also calculated Stone-Geisser's Q2 to evaluate the predictive capacity of the model and by obtaining all Q2 values greater than 0 we can conclude that it has a relevant predictive capacity [76]. See Table 6.

 Table 6: Predictive Summary of the latent variables

RMSE

MAE

Q²_predict

Behavioural Intention

0,803

0,616

0,363

Use

0,999

0,877

0,009

Thank you for your recommendation about terms of both the syntax and sentence construction. We have reviewed the entire document and made changes in several sentences in yellow in the reviewed manuscript.

Reviewer 2 Report

This work bases on many hypotheses and conducts statistical analysis then yields results and conclusions. It is a practical mode of researching but relies on formal inference or reasoning methods.

The descriptions are plain and clear. This manuscript is straightforward and clear to the conclusion.

Author Response

Firstly, thank you for your effort in reviewing our paper and, thank you for your kind words. No action is required.

Thank you for your recommendation about the English language and style. We have reviewed the entire document and made changes in several sentences, in yellow in the reviewed manuscript.

Reviewer 3 Report

The paper intends to analyze the the factors influencing the use of e-commerce on the basis of recommender systems

  1. which kind of recommender systems and their influence on e-commerce are analyzed? This question cannot be answered after reading the paper
  2. There is a quite big bias in the response group which was never quantified, based on the data
  3. The paper is related to AI in the way, that SOME recommender systems are AI based. Because we don't know, which RS had been analyzed, this paper stays on a very generic level in this purpose
  4. The requested integration of the use of Big Data and AI tools in the business management system (p.2) is not shown in the paper
  5. A sample size of 400 is far from the named Big Data in the paper
  6. Based on that the possibility of generalization of the results is for me under question until now.

Author Response

1. Thank you for your comment.

Initially, the questionnaire was designed with several AI-based recommendation systems in mind as we can see from the questionnaire header:

“The Artificial Intelligence can be a complex term of to define, although basically we could define it as the understanding and to build intelligent entities. There is Artificial Intelligence in many more things of the that we imagine and among them, we can highlight:

- Systems of recommendation. Every time that we buy something, for example, in Amazon they recommend us additional products according to our profile and previous purchases.

- Recommendations based on your position. If you are looking for something or a place, your phone recommends where to go or what to do.

- And many other applications....”

So in the second paragraph of the 3.2 epigraph, we have included in the text of the revised manuscript:

"In the heading of the questionnaire used we include the following explanation of what an ORS is: Every time that we buy something, for example, in Amazon, it makes additional products recommendations according to our profile and previous purchases; or if you are looking for something or a place, your phone recommends where to go or what to do."

2. The selected sample is made up of 448 questionnaires from individuals who have stated that they use online recommendation systems when faced with purchasing decisions. The analysis of socio-demographic variables in the sample indicated that 53,1% are female, the average age is 27,1 years old and 71,6% are single. Only 23.2 per cent are hired workers and 64.9% are students. 68.75per cent live at home and have monthly incomes over 1,500 €. More than half of the sample (50.7%) live in cities with more than 100,000 people.

We also sent the questionnaire with a convenience snowball sampling method and finally, we got 448 answers. We have highlighted that this can be a limitation but in Spain most users of these technologies are young so we believe that the sample can be a true reflection of current users. (see  Daniel Mican, Dan-Andrei Sitar-Tăut and Ovidiu-Ioan Moisescu, Decision Support Systems, https://doi.org/10.1016/j.dss.2020.113420)

3. Thank you for your recommendation.

Indeed. The idea of this research is to generate content that can be tested in different RSs that are AI-based, especially because this kind of system is in constant development. However, for the questionnaire, we used the examples in point 1 as a reference to respondents. This has been included in the modified text of the manuscript in section 3.2.

4. In the Practical implications sub-section we have included in the text of the revised manuscript:

"The fact that the two variables which have a greater influence on the behavioural intention of recommendation systems are performance expectancy and hedonic motivations, makes us recommend the developers of these systems to try to adapt them to the benefit expected by the online buyer, both at the efficiency level in the recommendations using Big Data Analytics [84] given in the purchasing process and at the level of design and usability, to attain a greater enjoyment when these systems appear in the online purchasing process."

5. Thank you for your comment, but our study does not propose to make recommendations with the big data, but about the use of big data. In this way and based on recommendations such as those of Henseler, Hubona and Ray, 2016; Hair et al., 2017, the sample of 448 individuals are adequate and sufficient to serve the purposes of this study.

6. We are aware that the sample size does not allow us to generalize conclusions, since the sample has a very specific profile and is not representative of the population. However, our study is new in this field and we offer a new and predictive model of the user's behaviour towards big data and AI that can be applied in other circumstances and with more representative samples.

Furthermore, in order to ensure the predictive capability of the model we have calculated Stone-Geisser's Q2 and added the following paragraph at the end of section 4:

"We have also calculated Stone-Geisser's Q2 to evaluate the predictive capacity of the model and by obtaining all Q2 values greater than 0 we can conclude that it has a relevant predictive capacity [76]. See Table 6.

Table 6: Predictive Summary of the latent variables

RMSE

MAE

Q²_predict

Behavioural Intention

0,803

0,616

0,363

Use

0,999

0,877

0,009